# Cathodic Water Enhances Seedling Emergence and Growth of Controlled Deteriorated Orthodox Seeds

**DOI:** 10.3390/plants10061170

**Published:** 2021-06-09

**Authors:** Kayode Fatokun, Richard P. Beckett, Boby Varghese, Norman W. Pammenter

**Affiliations:** 1School of Life Sciences, University of KwaZulu-Natal, Westville Campus, Private Bag X54001, Durban 4000, South Africa; varghese@ukzn.ac.za (B.V.); pammente@ukzn.ac.za (N.W.P.); 2School of Life Sciences, University of KwaZulu-Natal Pietermaritzburg, Private Bag X01, Pietermaritzburg 3209, South Africa; rpbeckett@gmail.com; 3Open Lab ‘Biomarker’, Kazan (Volga Region) Federal University, Kremlevskaya str. 18, 420008 Kazan, Russia

**Keywords:** cathodic water, deterioration, membrane, orthodox seeds, priming

## Abstract

All orthodox seeds eventually deteriorate during storage, a well-known problem in seed banking. Here we used a greenhouse study to test if priming deteriorated seeds with cathodic water can improve the emergence and subsequent seedling growth of three South African tree species, *Bolusanthus speciosus*, *Combretum erythrophyllum* and *Erythrina caffra.* Other priming solutions investigated were calcium magnesium (CaMg) solution and deionized water. In the present study, seeds were subjected to an artificial deterioration by increasing their water content to 14% and keeping them at 40 °C and 100% RH until they had lost 50% of their germination under laboratory conditions. Fresh and deteriorated seeds were primed with cathodic water, CaMg solution and deionized water, with non-primed fresh and deteriorated seeds as controls. Controlled deterioration significantly reduced total emergence and the biomass and photosynthetic parameters of the resulting seedlings. In one species (*Bolusanthus speciosus*), priming the deteriorated seeds with cathodic water significantly improved emergence parameters. However, in all species cathodic water significantly improved the total biomasses and other growth parameters of the seedlings derived from deteriorated seeds. Priming with CaMg solution and deionized water had little effect on emergence and while improving the growth of seedlings derived from deteriorated seeds, they were less effective than cathodic water. In fresh seeds, priming with all solutions resulted in small improvements in some parameters. Controlled deterioration of fresh seeds reduced the membrane stability index (MSI) in two of the three species and in all species increased the levels of the lipid oxidation products MDA and 4-HNE. Priming deteriorated seeds with cathodic water increased the MSI and reduced the MDA contents in all species and the 4-HNE content in one species. Other priming solutions were generally less effective in ameliorating oxidative stress. Results suggest that the strong antioxidative properties of cathodic water can explain its ability to ameliorate deterioration. In conclusion, the present study shows that priming with cathodic water is an effective way of invigorating deteriorated orthodox seeds and that it may have considerable potential in orthodox seed conservation.

## 1. Introduction

Orthodox seeds need to be stored in the short term so that they can produce high quality plants in the next growing season. Seeds also must be stored in long-term base collections with the aim of conserving genetic resources so that germplasm can be maintained and used in future breeding programs and for restoring wild populations [1]. Seed quality, therefore, must be maintained over extended periods. However, no matter how good the conditions, all seeds undergo deterioration (also known as aging) during long-term storage, which leads to a decline in seed germination, seedling emergence and later plant growth [2]. The reduction in “performance” is of considerable concern with respect to the long-term conservation of genetic diversity, both of wild species and in species important in the agricultural and horticultural sectors [1].

A major cause of aging is oxidative stress. During storage, seeds accumulate reactive oxygen species (ROS) such as superoxide (O_2_^−^), hydrogen peroxide (H_2_O_2_) and the hydroxyl radical (**·**OH) [3]. ROS attack many biomolecules and in particular the lipids in the cell membrane. Seeds have internal protective mechanisms such as antioxidant enzymes, including superoxide dismutase (SOD) and catalase (CAT), to counter the damaging effects of ROS. However, as deterioration progresses, the ability of these internal protective mechanisms becomes overwhelmed and damage can occur. Lipid peroxidation is believed to be a significant factor in seed deterioration and principal among the final products of lipid peroxidation are *4*-*hydroxynonenal* (4-HNE) and *malondialdehyde* (MDA) [4]. The presence of 4-HNE and MDA indicates that the cell membrane may have become permeable and that solutes may leak from the cell. Clear evidence now exists that losses in membrane integrity are a result of the uncontrolled production of ROS [5,6,7]. While seeds suffering oxidative stress will probably display reduced germination and emergence, in addition “hangover effects” of deterioration may be carried through to later growth stages of the plant. Seedlings of deteriorated seeds may display reduced photosynthesis and transpiration, slower growth and ultimately lower yields (for review see [8]). Thus, even if poor quality seeds germinate or emerge, the quality of plants generated from such seeds is not guaranteed.

Seed invigoration, although sometimes used interchangeably with seed priming, is an umbrella term of treatments that can improve the germination and field performance of a given seed lot [1]. It is basically a pre-sowing technique used to improve germination and seedling growth or to facilitate the performance of seeds the time of sowing [1,9]. Seed invigoration includes pre-sowing hydration treatments [10,11,12], low-molecular-weight osmo-protectant seed treatments [9], coating technologies [13] and more recently, pre-sowing dry heat treatment [11]. The focus of seed invigoration is to improve germination, reduce seedling emergence time and increase uniformity of emergence and protection of seeds from biotic and abiotic factors during the critical phases of seedling establishment.

There are essentially two ways of priming seeds. In the first, the hydrated seeds are sown after a period of hydration [14], reducing the lag time of imbibition [15,16]. Usually when seeds are not re-dried, they are placed in solutions with a high osmotic potential to prevent the seeds from taking in enough water to enter phase III (initiation of growth) of hydration. The notion is to hold the seeds in phase II (reactivation of metabolism) and therefore essentially keeping the seed within the lag phase [9]. During this period, the seeds are metabolically active and stored reserves are converted for use during germination. After imbibition, the seeds are removed from the priming solution, rinsed with water and sown in the field [14,17]. However, it is also possible to dry seeds before they are sown. In both methods, the priming hydration causes activation or repair of enzymes [17]. If seeds are dried after priming, it has been suggested that damaged seeds undergo repair during drying, possibly explaining the improved performance of primed seeds [17]. Numerous advantages of priming orthodox seeds have been reported for plants from around the world [18,19]. Primed seeds can exhibit faster germination when re-imbibed under normal or stress conditions. Priming has been shown to benefit a range of crops from tropical regions, for example *Oryza sativa*, *Zea mays*, *Sorghum bicolor* and *Cajanus cajan* [14,20].

In this study, we tested the ability of a novel seed priming agent, cathodic water, to ameliorate deterioration in three South African tree species, *Bolusanthus speciosus*, *Combretum erythrophyllum* and *Erythrina caffra*. Cathodic water is an electrolyzed form of calcium magnesium (CaMg) solution [21]. Cathodic water has strong antioxidative properties and in addition to the benefits conferred by conventional priming solutions, may counteract the damage caused by ROS. Our study therefore brings the concept of electrochemistry into plant germplasm conservation and crop establishment and production. Preliminary reports from our group showed that under laboratory conditions, priming with cathodic water had beneficial effects on the germination of deteriorated seeds from a variety of species [22,23]. However, its effects on seedling emergence and subsequent growth of the resulting seedlings have not been properly documented. Rather than use naturally aged seeds, here we used controlled deterioration, involving exposing seeds to a predetermined aggravated temperature and humidity [24,25]. This enabled us to deteriorate the seeds of all species to the same extent.

## 2. Materials and Methods

### 2.1. Study Area and Acquisition of Plant Materials

Seed aging and germination were carried out in the laboratories of the Plant Germplasm Conservation Research Unit, University of KwaZulu-Natal, Durban, South Africa. Plants were grown in a greenhouse of same School (Average temperature: 23.5 °C, relative humidity: 67%.

Potting mix and multifeed fertilizer used in this study were purchased from Silverhill seeds, Cape Town, South Africa. Initial germination tests carried out on the seeds indicated that both *B. speciosus* and *E. caffra* have seed coat imposed dormancy, while *C. erythrophyllum* also has a form of dormancy imposed by the samara covering the seed. Nicking was carried to break *E. caffra* and *B. speciosus* dormancy, while dormancy in *C. erythrophyllum* was broken by removing the samara covering the seeds. The initial water contents of the species were raised to 14% using vapor chambers. The seeds were then sealed in airtight glass jars and kept in a digital oven (Series 2000, Scientific, USA) at 40 °C and 100% relative humidity. Samples were taken every few days to assess the time needed to achieve a 50% reduction in germination.

### 2.2. Preparation of Cathodic Water

A solution containing 1 μM CaCl_2_ and 1 μM MgCl_2_ in deionized water known as calcium magnesium (CaMg) solution was prepared, autoclaved and stored at −5 °C until needed. The pH of CaMg solution was 7.

Cathodic water was prepared by electrolyzing the CaMg solution [26]. Two 200 mL glass beakers were filled with CaMg solution and platinum electrodes were immersed in the solution, the anode in one beaker and the cathode in the other. To form a complete circuit, an agar-based potassium chloride (KCl) salt bridge (a U shaped glass tube) was inserted to connect the two beakers. The CaMg solution was electrolyzed by applying a 60 V potential difference using a Bio-Rad Powerpac (BioRad, Hercules, CA, USA) at 400 mA for 1 h at room temperature. The electrolysis yielded anodic (oxidizing) water with a pH of approximately (*c*.) 2.4, and cathodic (reducing) water with a pH of *c*. 11.2. The anodic water was discarded while the cathodic water was used within one hour of preparation.

### 2.3. Seed Priming, Plant Management and Data Collection

To prime the seeds (50 seeds per treatment) they were placed between 20 layers of single-ply paper towel, which was placed on aluminum foil. Priming solutions (50 mL) were poured onto these paper towels. The aluminum foil containing the seeds was placed in plastic pouches and after 24 h, just before radicle emergence, the seeds were dried back to their original masses under ambient laboratory conditions for 7 d and kept at 4 °C in air-tight bottles until required. There were eight treatments: six seed priming treatments and two controls. The seed priming treatments were: aged seeds primed with cathodic water (ASP.CW); aged seeds primed with CaMg solution (ASP.CM); aged seeds primed with distilled water (ASP.DW); fresh seed (unaged) primed with cathodic water (FSP.CW); fresh seeds primed with CaMg solution (FSP.CM); fresh seeds primed with distilled water (FSP.DW) and two controls; unprimed fresh seeds (FSC) and unprimed aged seeds (ASC). Each treatment was replicated four times (*n* = 200). Note: *n* = total number of seeds used per treatment.

To test emergence and subsequent seedling growth, a completely randomized experimental design was used. The plants were grown in 2 L pots containing 800 g of peat-based potting mix and watered as required. After CD and priming, five seeds were planted in each pot, with four pots per treatment (*n* = 20) arranged in a completely randomized design. Emerged seedlings were counted and recorded daily. Thinning was done at 4 weeks after planting to reduce the number of plants per pot from five to one. Plants were irrigated when required throughout the period of the experiment. Grovida multifeed water soluble fertilizer was used to supply nutrients to the growing plants at 1 g L^−1^ weekly The composition of the fertilizer was N, P, K, S and Mg at 193, 83, 153, 6.1 and 4.6 g kg^−1^, respectively. The fertilizer also contained the micronutrients Zn, B, Mo, Fe, Mn and Cu at 700, 1054, 63, 751, 273 and 75 mg kg^−1^, respectively. Physiological parameters (photosynthesis, transpiration and chlorophyll fluorescence) were measured 8 weeks after planting (WAP) and chlorophyll content 10 weeks after planting (WAP). The study was terminated at 12 WAP. The plants were harvested and separated into root, stem and leaves. The plant parts were measured and the leaves were counted. All plant parts were oven dried at 65 °C to constant mass. The oven-dried plant parts were weighed and recorded.

#### 2.3.1. Emergence

Seedling emergence counts were taken daily until constant counts were achieved. The value was expressed in percentage. The following were determined from the emergence data taken: First Day of Emergence (FDE), Final Emergence Percentage (FEP), Mean Emergence Time (MET). MET was calculated according to the equation of [27]: MET = ∑Dn/*n*, where *n* is the number of emerged seedlings on day D and D is the number of days counted from the beginning of seedling emergence.

Emergence Index (EI) is also called speed of emergence and it was calculated using the method of [28].
$$=\frac{\mathrm{Number}\;\mathrm{of}\;\mathrm{emerged}\;\mathrm{seedlings}}{\mathrm{Days}\;\mathrm{of}\;\mathrm{first}\;\mathrm{count}}+\ldots +\frac{\mathrm{Number}\;\mathrm{of}\;\mathrm{emerged}\;\mathrm{seedlings}}{\mathrm{Days}\;\mathrm{of}\;\mathrm{final}\;\mathrm{count}}$$

Uniformity of Emergence (UE) was calculated using the formula of [29], =∑*n*/[(T − t)^2^*n*] (*n* is the number of emerged seedlings counted on a particular day, t is the time (number of days) beginning from day 0, T is the MET.

#### 2.3.2. Leaf Chlorophyll Content and Chlorophyll Fluorescence

Leaf chlorophyll content was measured on the third, fourth and fifth leaves counting from the terminal bud using a SPAD chlorophyll meter (model SPAD-502; Minolta Corp., Ramsey, NJ, USA). Three measurements were taken on each leaf at 10 weeks of growth. The chlorophyll content was estimated as the mean of the 9 readings and expressed as the chlorophyll content index (CCI).

Chlorophyll fluorescence was measured using a Li-Cor 6400XT portable photosynthesis measuring system (Li-Cor, Lincoln, NE, USA). Chlorophyll fluorescence transients were measured on the third leaf from the terminal bud across all treatments and replicates at 8 weeks after planting. Measurements were taken after the plants were dark adapted for 40 min. Fv/Fm, the ratio of variable (Fv) to maximum fluorescence (Fm), was used as a measure of potential photochemical efficiency of photosystem II (PSII).

#### 2.3.3. Photosynthetic Capacity—Steady-State Gas Exchange

Gas exchange was measured with a Li-Cor 6400 portable photosynthesis measuring system, fitted with a standard chamber and configured as an open system (Li-Cor, Lincoln, NE, USA). Measurements were taken at 8 weeks after planting across all treatments and replicates. Instantaneous measurement of leaf-based CO_2_ assimilation and transpiration rates were carried out at a CO_2_ concentration of 400 µmol CO_2_ mol^−1^, a light intensity of 1000 µmol m^−2^ s^−1^ and a temperature of 25 °C; measurements were taken between 11:00 am to 2:00 pm when conditions were typically most stable. The measurements were taken on fully expanded, non-senescing leaves. Three measurements were taken per plant on the third, fourth and fifth leaves from the terminal bud. The mean of these three measurements was used at the average rate of photosynthesis for each plant.

#### 2.3.4. Harvesting and Post-Harvest Data Collection

Shoot heights were measured using a meter rule at 12 WAP and then plants were carefully pulled out of the potting mix to avoid damage to the roots. Potting mix which adhered to the roots was removed using tap water. The lengths of the roots were measured, the number of leaves was counted and the leaf area was measured with a leaf area meter (CI-202 Area Meter, CID, Inc., Camas, WA, USA). The plants were subsequently separated into leaves, stems and roots. The plant parts (root, stem and leaves) were oven dried at 65 °C until constant mass and then weighed.

### 2.4. Determination of MDA and 4HNE Contents in Seeds

In a separate experiment, fresh and deteriorated seeds were hydrated in the range of solutions as described above and at about the end of phase II of the seed hydration (18 h (*B. speciosus*) and 20 h (*C. erythrophyllum* and *E. caffra*)). One gram of seeds was homogenized in 5 mL of 20% (*w*/*v*) trichloro acetic acid (TCA) consisting of 0.5% (*w*/*v*) thiobarbituric acid (TBA). The homogenate was then incubated for 30 min at 95 °C [30], after which it was placed in an ice bath for 10 min and thereafter centrifuged at 10,000× *g* for 10 min. The absorbance of the supernatant was read at 600 nm using PowerWave™ microplate spectrophotometer (BioTek Instruments, Inc, Winooski, VT, USA). The content of MDA was calculated using the extinction coefficient of 155 mM^−1^ cm^−1^. Content of MDA was expressed as mmol g^−1^ fresh mass.

Similarly, a separate batch of fresh and deteriorated seeds was hydrated in the range of solutions described above and the content of 4-HNE was estimated at about the end of phase II of hydration. Seeds (1 g) were homogenized in 5 mL of cold borate buffer (0.2 M, pH 7.4) at 4 °C. The homogenate was then mixed with 10% (*w*/*v*) TCA and centrifuged at 12,000× *g* for 15 min. The supernatant obtained was thoroughly mixed with 2, 4-dinitrophenyl hydrazine (1 mg mL^−1^ in 0.5 M HCl). The complex obtained was kept at laboratory conditions for 2 h after which it was extracted in hexane and evaporated under liquid nitrogen. The residue was dissolved in methanol and absorbance was read at 350 nm against methanol as blank [31]. Content of 4-HNE in the samples was expressed as mmol g^−1^ fresh mass.

### 2.5. Determination of Seeds Membrane Stability Index

Membrane stability index of the seeds was assessed using the method of [32]. MilliQ water (MW) (Millipore, Gradient A-10, Burlington, MA, USA) (20 mL) was added to two test tubes and seeds (1 g) were added to each tube. One tube was placed in a water bath at 40 °C for 40 min (T1) and the other in a water bath at 100 °C for 15 min (T2). Electrical conductance of the water in both test tubes after incubation was measured with a multi-cell electrical conductivity meter (Reid and Associates, Durban, South Africa). The measurement was repeated four times and the average of the four trials was reported as the electrical conductance and used in calculating the membrane stability index (MSI) of the seeds; MSI = [1 − (T1/T2)] × 100.

### 2.6. Statistical Analysis

The data collected were subjected to analysis of variance (ANOVA) using GenStat Release 12.1 (PC/Windows Vista) (VSN International Ltd., Hemel Hempstead, UK, 2009). Means of the treatments were compared using the Tukey test at 5% least significant difference (LSD_0.05_).

## 3. Results

### 3.1. Effect of Cathodic Water on Seedling Emergence

Seedling emergence was delayed as a result of controlled deterioration of seeds in all the test species. Emergence was delayed for about 2 d in *E. caffra* and *B. speciosus* and 3 d in *C. erythrophyllum* (Table 1). Total seedling emergence, mean emergence time, emergence index and uniformity of emergence were also adversely affected by controlled deterioration when compared with the fresh unprimed seeds. Priming deteriorated seeds promoted earlier and more uniform emergence when compared with the control. While for *B. speciosus* the effects were mostly significant for cathodic water, the other priming solutions had smaller effects that were often not significant. In *C. erythrophyllum*, while priming tended to improve emergence, the effects were mostly not significant. In contrast, priming fresh seeds only resulted in small increases in total emergence in all species.

### 3.2. Effect of Priming on Seedling Growth

Priming fresh seeds with all solutions tended to increased growth parameters (Table 2 and Table 3). In general, cathodic water was most effective, although the differences between cathodic water and the other solutions were not always significant. The increases in total dry mass were significant for all priming solutions for *B. speciosus* and *C. erythrophyllum*, but only for cathodic water for *E. caffra*. Controlled deterioration of seeds significantly reduced the subsequent growth of plants derived from the deteriorated seeds. For example, root length was typically reduced by *c*. 50% (Table 2). The numbers of leaves were reduced by 23% in *C. erythrophyllum*, 35% in *B speciosus* and 43% in *E. caffra* (Table 2). The biomasses of the individual plant parts and the total biomasses of all species were very significantly reduced in seedlings derived from deteriorated seeds (Table 3). Invigorating deteriorated seeds with any of the priming solutions greatly increased the growth parameters of all species. Most of the improvements were significant for the seeds primed with cathodic water, while the improvement in the seeds primed with CaMg solution and deionized water treatments were smaller and not always significant (Table 3). In some cases, the parameters were more than double compared with plants derived from unprimed aged seeds.

### 3.3. Effect of Priming on Photosynthetic Parameters

For all species, seedlings derived from deteriorated rather than fresh seeds had significantly lower chlorophyll contents, photochemical efficiencies and rates of photosynthesis and transpiration (Figure 1 and Figure 2). While having little effect on fresh seeds, priming tended to increase these parameters in deteriorated seeds. In *B. speciosus*, cathodic water significantly improved these parameters, while the effects of the other priming solutions tended to be less and were not always significant. In the other two species, the effects of priming were smaller and usually not significant, although in general, cathodic water gave the best results.

### 3.4. Effect of Priming on Membrane Stability and the Levels of Oxidized Lipids

In fresh seeds, priming tending to increase the MSI, significantly for cathodic water and distilled water in *B. speciosus* and for cathodic water in *C. erythrophyllum*. Primed fresh seeds had significantly lower MDA levels than unprimed seeds in *B. speciosus* and *C*. *erythrophyllum*, with cathodic water reducing MDA levels significantly more than the other priming solutions. *E. caffra* priming had no effect on MDA levels in fresh seeds. Priming fresh seeds with cathodic water significantly reduced 4-HNE in all species, but the other priming solutions had no significant effect. Controlled deterioration of the seeds of *C. erythrophyllum* and *E. caffra* significantly reduced their MSI compared with fresh seeds, but the reduction was smaller and not significant in *B. speciosus* (Figure 3). For all species, deterioration significantly increased the levels of the lipid oxidation products MDA and 4-HNE (Figure 4). The MSI of deteriorated seeds primed with all solutions had significantly better MSI than unprimed seeds, except in *B. speciosus* where only cathodic water had a significant effect (Figure 3). All priming solutions significantly reduced the amounts of lipid peroxidation product MDA in the seeds and cathodic water was significantly more effective than the other priming solutions for *C. erythrophyllum* and *B. speciosus* (Figure 4). However, priming only reduced 4-HNE levels in *E. caffra* and all solutions were equally effective.

## 4. Discussion

The main conclusion of the work presented here is that for all three of the South African tree species tested, priming seeds with the strong reducing agent cathodic water significantly improved the growth of seedlings derived from deteriorated seeds. Furthermore, in one of the species, *B. speciosus*, the emergence of deteriorated orthodox seeds was also significantly increased by priming with cathodic water. While our earlier laboratory studies showed that cathodic water can improve the germination of deteriorated seeds [22,23], here we show that the benefits extend to emergence and the later stages of seedling development. The effects of cathodic water on the levels of lipid peroxidation products are generally consistent with the view that cathodic water acts by reducing the levels of ROS in aged seeds. We suggest that priming with cathodic water can be recommended as a useful tool to improve the conservation of orthodox seeds.

### 4.1. Effects of Priming on Emergence

In all species, controlled deterioration delayed seedling emergence and reduced the uniformity of emergence and the emergence index (Table 1). The first day of emergence was delayed by 3 d in *C. erythrophyllum* and about 2 d in *E. caffra* and *B. speciosus* In *B. speciosus* priming deteriorated seeds promoted earlier and more uniform emergence. The effects were mostly significant for cathodic water, while the other priming solutions had smaller effects that were often not significant. In the other two species, while priming tended to improve the emergence parameters, the effects were smaller and mostly not significant. Although the effects of cathodic water on emergence have not been tested before, priming with other solutions has been shown to improve emergence parameters. For example, priming was reported to promote the earlier emergence of seedlings in *Oryza sativa* [33] and *Zea mays* [34]. A particularly important emergence parameter is uniformity, because uniformity can improve stand establishment and increase biomass, particularly when conditions are suboptimal, such as during drought, salinity and water stress [33,34]. Uniformity of emergence can also reduce weed-inflicted yield loss, for example by up to 10% in rice [35]. While the improvements in emergence in *C. erythrophyllum* and *E. caffra* were too small to be significant, priming with cathodic water clearly significantly improves emergence in *B. speciosus*.

### 4.2. Effect of Priming on Seedling Growth

Priming fresh seeds with cathodic water and other solutions resulted in small, and in most cases not significant, increases in the growth of the resulting seedlings (Table 2 and Table 3). The small improvements that occurred may be due to the repair of the natural deterioration that occurred before harvesting and during seed storage [36,37]. Seedlings derived from seeds subjected to controlled deterioration displayed significantly reduced growth (Table 2 and Table 3). Invigoration of deteriorated seeds with all priming solutions in water resulted in better seedling growth for all species (Table 2 and Table 3). Most of the improvements were significant for the seeds primed with cathodic water, while the improvement in the seeds primed with the other priming solutions were smaller and often not significant. Interestingly, while priming only improved emergence in one of the species tested here (*B. speciosus*, Table 1), cathodic water significantly improved the growth parameters of seedlings derived from deteriorated seeds in all species tested here. The improvement in seedling growth in plants derived from primed seeds may have occurred due to increases in the activities of enzymes such as *α-amylase*. Such increases in enzymatic activities have been reported to promote the hydrolysis of starch into soluble sugars for seed respiration and better growth [22]. While seed priming is known to promote seedling growth (for review see [38]), cathodic water is clearly more effective that the other solutions tested here.

### 4.3. Effects of Priming on Photosynthetic Parameters

One of the ways that priming improves seedling growth is by improving photosynthesis. In all species, seed deterioration significantly reduced the chlorophyll contents, photochemical efficiencies and rates of photosynthesis and transpiration in the resulting seedlings (Figure 1 and Figure 2). While having little effect on fresh seeds, priming tended to increase these parameters in deteriorated seeds. In *B. speciosus*, all effects of cathodic water were significant, while the effects of the other priming solutions tended to be less and were not always significant. In the other two species, the effects of priming were smaller and usually not significant, although in general, cathodic water gave the best results. Although not tested for cathodic water, priming with other solutions is well known to improve photosynthesis in the resulting seedlings [39] (for review see [38]). Theoretically, priming may have increased the capacity for photosynthetic electron transport, or possibly a greater investment in enzymes of the Calvin cycle [40]. While we did not study the mechanisms of the improvement in photosynthesis in detail, the net result of the general improvements in photosynthesis was an increase in plant growth (Table 2 and Table 3)

### 4.4. Effects on Membrane Leakage and Lipid Peroxidation Products

The effects of priming on membrane leakage and lipid peroxidation, particularly with cathodic water, are generally consistent with priming reducing oxidative stress in the seeds of the three species tested here. As discussed in the Introduction, a gradual increase in oxidative stress is considered to be a major reason for the deterioration of seeds during aging. Controlled deterioration significantly reduced the MSI of *C. erythrophyllum* and *E. caffra*, but not *B. speciosus* (Figure 3). For all species, deterioration significantly increased the levels of the lipid oxidation products MDA and 4-HNE (Figure 4). All priming solutions significantly increased the MSI, except in *B. speciosus* where only cathodic water had a significant effect (Figure 3). All priming solutions significantly reduced the amounts of lipid peroxidation product MDA in the seeds and cathodic water was significantly more effective than the other priming solutions for *C. erythrophyllum* and *E. caffra* (Figure 4). The greater efficacy of cathodic water may be linked to its strong antioxidant property. However, priming only reduced levels of the other oxidation product we tested, 4-HNE, in *E. caffra* and all solutions were equally effective. Lipid oxidation and MSI are intimately linked, as lipid oxidation is well known to increase membrane permeability [41]. Membrane permeability leads to leakage of solute from the cells leading to loss of critical nutrients such as sugar and amino acids and consequently reduced growth. The differences observed among the species in regards to the impact of controlled deterioration as evidenced in the changes that occurred in MDA, 4-HNE and MSI essentially may be due to differences in species biology (not investigated in this study).

There have been numerous studies that have shown that priming reduces oxidative stress in seeds (for review see [42]). It is possible that priming upregulates ROS scavenging enzymes in the seeds. For example, References [43,44] report that priming seeds of *Spinacia oleracea* upregulates APX, SOD, CAT and GR. Reference [45] showed that during the aging of *Helianthus annuus* seeds, H_2_O_2_ accumulated and the expression of CAT was reduced. Priming both invigorated the seeds and increased the expression and activity of CAT. Alternatively, or perhaps additionally, the priming-induced reductions in lipid oxidation products and increases in MSI may have been because priming more generally reduced ROS formation [46]. In the present study, cathodic water may have directly reacted with ROS. Recently, Reference [47] showed that treating excised embryonic axes from the recalcitrant seeds of the South African tree *Ekebergia capensis* with cathodic water both improved survival of the explants following cryopreservation and also reduced the formation of ROS. However, as other priming solutions were also beneficial, in addition to direct reactions with ROS, cathodic water may have reduced ROS levels in other ways e.g., by improving the activity of ROS-scavenging enzymes. The precise mechanisms whereby priming increased MSI and reduced lipid peroxidation were not investigated in the present study. While priming with all solutions reduced oxidative stress and improved the MSI, cathodic water tended to have a slighter stronger ameliorative effect, particularly in its ability to reduce MDA formation (Figure 4). This may explain why the strongly reducing cathodic water was better overall at improving the growth of seedlings derived from deteriorated seeds.

## 5. Conclusions

Our original motivation for testing the effectiveness of cathodic water in germplasm conservation came from an earlier study, in which the deterioration of maize seed was counteracted by cathodic protection [48]. In that study, maize seeds were placed on an aluminum foil disc with the disc attached to the cathode of a power pack [48,49]. However, a solution of cathodic water is more practical to use than foil discs and in later experiments we showed that cathodic water can improve the cryopreservation of the embryonic axes and embryos of recalcitrant seeds [21,47,50] and shoot tips [51], and could improve the germination of orthodox seeds [22,23]. In the work presented here we showed that cathodic water is a particularly effective priming agent for improving the emergence and subsequent seedling growth of deteriorated seeds. While all priming solutions were capable of some measure of invigoration, priming with cathodic water was most effective. Results from measuring membrane leakage and lipid oxidation in primed seeds are generally consistent with the view that the benefits of cathodic water derive from its ability to reduce oxidative stress, probably at least in part because of its strong antioxidant capacity. The deterioration of seeds in seed banks is of global concern, as it affects the long-term conservation of genetic diversity of both wild species and agricultural plants [52] essential for future breeding programs. In the future, it will be necessary to produce varieties that perform well under future climate change scenarios, particularly in sub-Saharan Africa where the effects of climate change are likely to be severe [53,54]. Results presented here show that priming deteriorated seeds with cathodic water can be an effective means of improving orthodox seed conservation.

## Figures and Tables

**Figure 1 plants-10-01170-f001:**
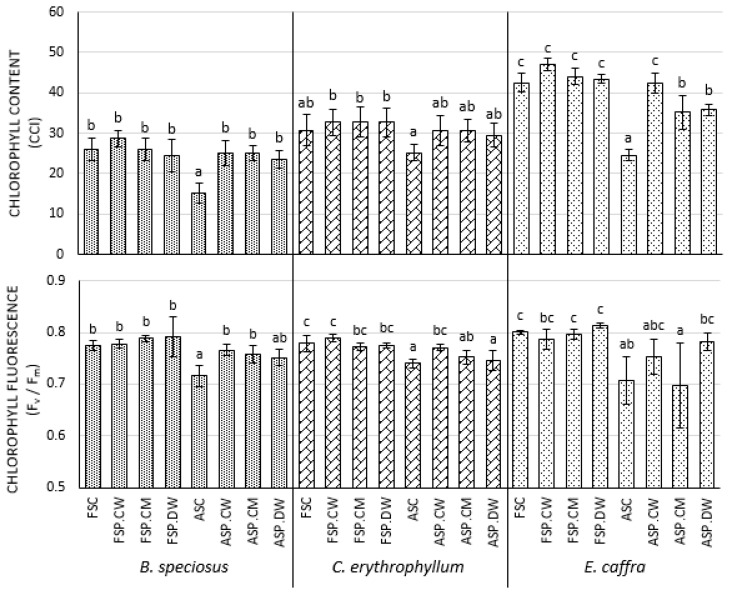
Effect of cathodic water, calcium magnesium solution and deionized water seed invigoration on the chlorophyll fluorescence and chlorophyll content in the leaves of *Bolusanthus speciosus*, *Combretum erythrophyllum* and *Erythrina caffra*. Bars with different letters in each species and for each parameter (chlorophyll content/ chlorophyll fluorescence) investigated are significantly different (*p* < 0.05). FSC = fresh seeds that were neither aged nor primed; ASC = seeds that were aged but not primed; FSP.CW = fresh seeds primed with cathodic water; FSP.CM = fresh seeds primed with calcium magnesium solution; FSP.DW = fresh seeds primed with deionized water; ASP.CW = aged seeds primed with cathodic water; ASP.CM = aged seeds primed with calcium magnesium solution; ASP.DW = aged seeds primed with deionized water.

**Figure 2 plants-10-01170-f002:**
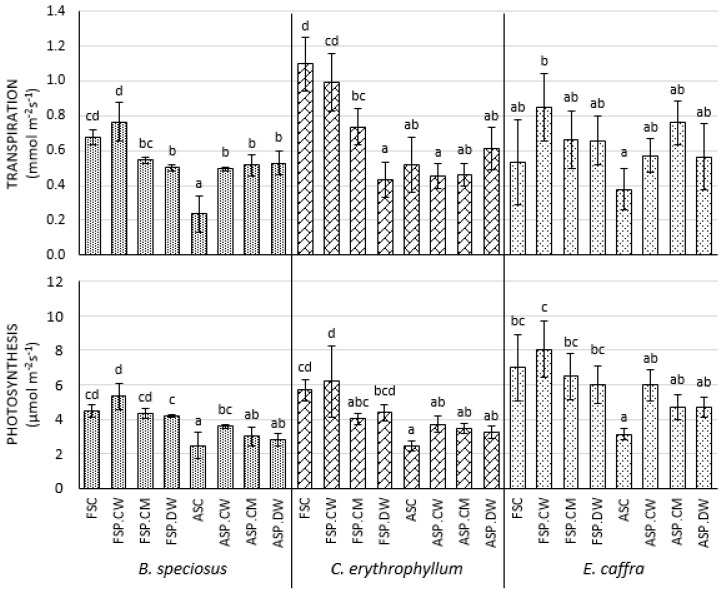
Effect of cathodic water, calcium magnesium solution and deionized water seed invigoration on the photosynthesis of *Bolusanthus speciosus, Combretum erythrophyllum* and *Erythrina caffra*. Bars with different letters in each species and for each parameter (photosynthesis/transpiration) investigated are significantly different (*p* < 0.05). Abbreviations as for Figure 1.

**Figure 3 plants-10-01170-f003:**
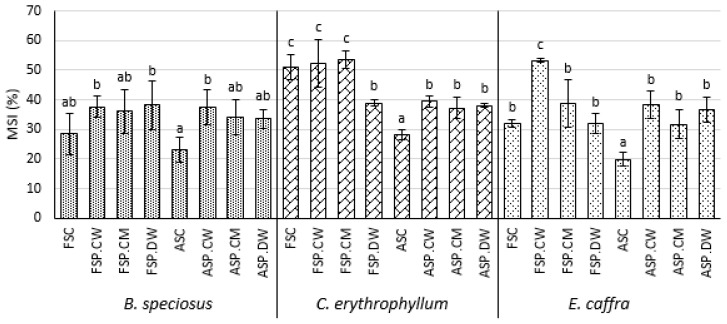
Effect of cathodic water, calcium magnesium solution and deionized water seed invigoration on the membrane stability index (MSI) of *Bolusanthus speciosus, Combretum erythrophyllum* and *Erythrina caffra*. Bars with different letters in each species are significantly different (*p* < 0.05). Abbreviations as for Figure 1.

**Figure 4 plants-10-01170-f004:**
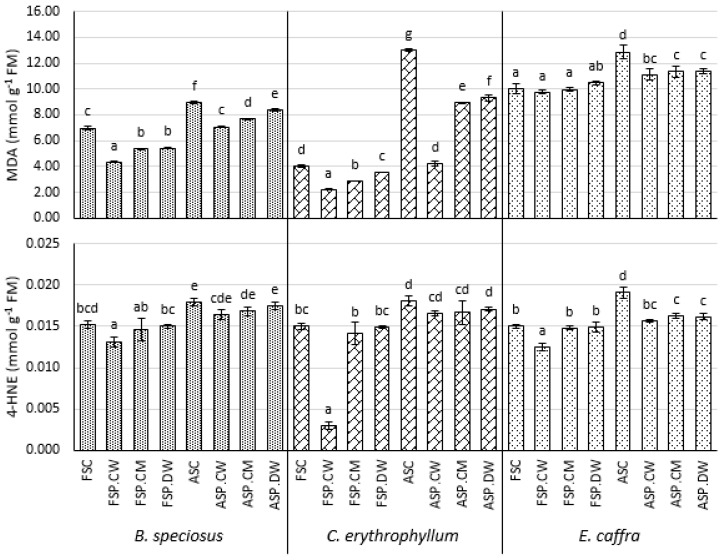
Effect of cathodic water, calcium magnesium solution and deionized water seed invigoration on MDA and 4-HNE contents in the seeds of *Bolusanthus speciosus, Combretum erythrophyllum* and *Erythrina caffra*. Bars with different letters in each species and for each parameter (MDA/4-HNE, 4-hydroxynonenal and malondialdehyde respectively) investigated are significantly different (*p* < 0.05). Abbreviations as for Figure 1.

**Table 1 plants-10-01170-t001:** Effects of cathodic water, calcium magnesium solution and deionized water treatments on the emergence of *Bolusanthus speciosus, Combretum erythrophyllum* and *Erythrina caffra*.

Emergence		FSC	FSP.CW	FSP.CM	FSP.DW	ASC	ASP.CW	ASP.CM	ASP.DW	LSD
First day of emergence	*B. speciosus*	6.0 ± 0.8 ^abc^	4.5 ± 0.5 ^a^	5.0 ± 0.6 ^ab^	7.0 ± 0.6 ^abcd^	9.5 ± 1.0 ^d^	7.0 ± 1.0 ^abcd^	8.0 ± 0.8 ^bcd^	8.5 ± 0.5 ^cd^	2.2 ± 1.05
Emergence%		85.0 ± 5.0 ^cd^	100.0 ± 0.0 ^d^	90.0 ± 5.8 ^d^	90.0 ± 5.8 ^d^	30.0 ± 5.8 ^a^	60.0 ± 8.2 ^bc^	50.0 ± 5.8 ^ab^	50.0 ± 5.8 ^ab^	16.6 ± 8.04
Mean emergence time		3.3 ± 0.1 ^c^	4.6 ± 0.041 ^d^	3.6 ± 0.2 ^c^	3.3 ± 0.2 ^c^	0.7 ± 0.2 ^a^	2.3 ± 0.3 ^b^	1.5 ± 0.3 ^ab^	1.6 ± 0.2 ^ab^	0.6 ± 0.30
Emergence index		1.6 ± 0.1 ^d^	2.6 ± 0.1 ^e^	1.7 ± 0.2 ^d^	1.4 ± 0.1 ^cd^	0.3 ± 0.1 ^a^	1.0 ± 0.1 ^bc^	0.6 ± 0.1 ^ab^	0.6 ± 0.1 ^ab^	0.3 ± 0.16
Uniformity of emergence		0.023 ± 0.001 ^d^	0.037 ± 0.001 ^e^	0.024 ± 0.001 ^d^	0.021 ± 0.002 ^cd^	0.010 ± 0.001 ^a^	0.016 ± 0.001 ^bc^	0.012 ± 0.002 ^ab^	0.012 ± 0.001 ^ab^	0.004 ± 0.002
First day of emergence	*C. erythrophyllum*	12.5 ± 0.5 ^a^	11.5 ± 0.3 ^a^	12.8 ± 0.6 ^a^	11.8 ± 0.5 ^a^	15.8 ± 1.1 ^b^	13.0 ± 0.6 ^ab^	13.0 ± 0.4 ^ab^	12.8 ± 0.6 ^a^	1.8 ± 0.88
Emergence%		60.0 ± 8.2 ^bc^	70.0 ± 5.8 ^c^	60.0 ± 8.2 ^bc^	45.0 ± 9.6 ^abc^	25.0 ± 5.0 ^a^	45.0 ± 5.0 ^abc^	35.0 ± 5.0 ^ab^	30.0 ± 5.8 ^ab^	19.6 ± 9.57
Mean emergence time		2.3 ± 0.3 ^bc^	2.8 ± 0.3 ^c^	2.5 ± 0.4 ^bc^	1.9 ± 0.5 ^abc^	0.7 ± 0.229 ^a^	1.5 ± 0.2 ^abc^	1.3 ± 0.2 ^abc^	1.2 ± 0.3 ^ab^	0.9 ± 0.43
Emergence index		1.2 ± 0.2 ^bc^	1.5 ± 0.2 ^c^	1.3 ± 0.2 ^bc^	1.1 ± 0.3 ^abc^	0.3 ± 0.111 ^a^	0.7 ± 0.095 ^abc^	0.7 ± 0.128 ^abc^	0.6 ± 0.144 ^ab^	0.5 ± 0.25
Uniformity of emergence		0.005 ± 0.0002 ^b^	0.006 ± 0.0003 ^b^	0.005 ± 0.0004 ^b^	0.005 ± 0.0005 ^b^	0.004 ± 0.0003 ^a^	0.004 ± 0.0002 ^ab^	0.004 ± 0.0002 ^ab^	0.005 ± 0.0002 ^ab^	0.001 ± 0.0004
First day of emergence	*E. caffra*	5.3 ± 0.3 ^ab^	4.5 ± 0.3 ^a^	4.8 ± 0.3 ^ab^	4.8 ± 0.3 ^ab^	6.8 ± 0.3 ^c^	5.8 ± 0.3 ^bc^	6.5 ± 0.3 ^c^	6.8 ± 0.3 ^c^	0.8 ± 0.37
Emergence%		100.0 ± 0.0 ^b^	100.0 ± 0.0 ^b^	100.0 ± 0.0 ^b^	100.0 ± 0.0 ^b^	40.0 ± 0.0 ^a^	50.0 ± 5.8 ^a^	45.0 ± 5.0 ^a^	50.0 ± 5.8 ^a^	9.9 ± 4.79
Mean emergence time		3.6 ± 0.1 ^b^	4.6 ± 0.1 ^c^	4.2 ± 0.2 ^bc^	4.1 ± 0.1 ^bc^	1.0 ± 0.074 ^a^	1.5 ± 0.1 ^a^	1.4 ± 0.2 ^a^	1.3 ± 0.2 ^a^	0.4 ± 019
Emergence index		2.5 ± 0.1 ^b^	4.1 ± 0.2 ^d^	3.3 ± 0.3 ^c^	3.2 ± 0.2 ^bc^	0.6 ± 0.060 ^a^	1.0 ± 0.092 ^a^	0.9 ± 0.139 ^a^	0.8 ± 0.119 ^a^	0.4 ± 0.22
Uniformity of emergence		0.059 ± 0.004 ^b^	0.136 ± 0.012 ^d^	0.095 ± 0.013 ^c^	0.086 ± 0.007 ^bc^	0.020 ± 0.001 ^a^	0.025 ± 0.001 ^a^	0.023 ± 0.002 ^a^	0.022 ± 0.002 ^a^	0.02 ± 0.01

FSC = fresh seeds that were neither aged nor primed; ASC = seeds that were aged but not primed; FSP.CW = fresh seeds primed with cathodic water; FSP.CM = fresh seeds primed with calcium magnesium solution; FSP.DW = fresh seeds primed with deionized water; ASP.CW = aged seeds primed with cathodic water; ASP.CM = aged seeds primed with calcium magnesium solution; ASP.DW = aged seeds primed with deionized water. ANOVA was performed across treatments. Means of replicate were separated at LSD_0.05_. Post hoc was done using a Tukey test. Means along the same row with different letters were significantly different (*p* < 0.05, *n* = 32).

**Table 2 plants-10-01170-t002:** Effects of cathodic water, calcium magnesium solution and deionized water treatments on the root length, stem length, number of leaves and leaf area of *Bolusanthus speciosus, Combretum erythrophyllum* and *Erythrina caffra*.

Treatment	*Bolusanthus speciosus*				*Combretum erythrophyllum*				*Erythrina caffra*			
	Root Length (cm)	Stem Length (cm)	Number of Leaves	Leaf Area (cm^2^)	Root Length (cm)	Stem Length (cm)	Number of Leaves (cm)	Leaf Area (cm^2^)	Root Length (cm)	Stem Length (cm)	Number of Leaves	Leaf Area (cm^2^)
FSC	34.3 ± 0.6 ^d^	13.0 ± 0.8 ^c^	13.0 ± 0.7 ^bc^	58.3 ± 4.2 ^bc^	36.0 ± 1.1 ^bcd^	16.5 ± 1.7 ^c^	15.0 ± 1.1 ^ab^	59.0 ± 1.1 ^d^	25.8 ± 1.1 ^c^	17.5 ± 1.0 ^cde^	12.8 ± 0.5 ^b^	716.2 ± 15.4 ^d^
FSP.CW	33 ± 1.1 ^cd^	12.8 ± 0.3 ^c^	16.0 ± 0.7 ^d^	112.4 ± 5.3 ^e^	41.3 ± 0.3 ^d^	21.5 ± 1.3 ^d^	22.5 ± 0.6 ^c^	78.8 ± 0.6 ^f^	28.5 ± 0.6 ^cd^	20.9 ± 1.4 ^e^	16.0 ± 0.4 ^c^	779.4 ± 17.2 ^d^
FSP.CM	33 ± 0.7 ^cd^	14.0 ± 0.7 ^c^	12.0 ± 0.4 ^bc^	72.6 ± 4.2 ^cd^	32.8 ± 1.0 ^bc^	16.6 ± 0.7 ^c^	16.5 ± 0.6 ^b^	77.9 ± 2.0 ^f^	24.8 ± 0.5 ^c^	17.0 ± 0.7 ^cd^	13.8 ± 0.3 ^bc^	739.8 ± 26.0 ^d^
FSP.DW	36.5 ± 2 ^d^	14.3 ± 0.2 ^c^	13.8 ± 0.6 ^cd^	74.1 ± 2 ^d^	36.3 ± 1.4 ^cd^	12.5 ± 0.6 ^bc^	15.3 ± 0.3 ^ab^	69.3 ± 1.4 ^e^	30.5 ± 1.0 ^d^	19.3 ± 0.6 ^de^	15.0 ± 0.4 ^bc^	736.2 ± 14.6 ^d^
ASC	18.3 ± 1.1 ^a^	7.5 ± 0.2 ^a^	8.5 ± 0.6 ^a^	21.2 ± 1.3 ^a^	18.5 ± 0.6 ^a^	7.2 ± 0.2 ^a^	11.5 ± 0.6 ^a^	20.5 ± 0.5 ^a^	11.9 ± 0.9 ^a^	9.9 ± 0.4 ^a^	7.3 ± 0.5 ^a^	86.3 ± 0.6 ^a^
ASP.CW	31.3 ± 1.3 ^bcd^	12.5 ± 1.2 ^c^	11.5 ± 0.6 ^bc^	58.0 ± 1.8 ^bc^	30.8 ± 0.5 ^b^	12.1 ± 1.1 ^bc^	14.3 ± 0.8 ^ab^	45.9 ± 1.7 ^c^	26.3 ± 0.6 ^c^	14.3 ± 0.5 ^bc^	13.5 ± 0.6 ^b^	642.7 ± 9.8 ^c^
ASP.CM	28.5 ± 1.2 ^bc^	9.0 ± 0.9 ^ab^	10.5 ± 0.3 ^ab^	31.0 ± 2.1 ^a^	21.9 ± 2.2 ^a^	11.4 ± 0.7 ^ab^	14.2 ± 0.8 ^ab^	27.0 ± 1.7 ^b^	18.8 ± 1.0 ^b^	12.6 ± 0.6 ^ab^	8.5 ± 0.6 ^a^	122.7 ± 1.2 ^a^
ASP.DW	25.7 ± 1.2 ^b^	12.0 ± 0.4 ^bc^	8 ± 0.4 ^a^	50.5 ± 2 ^b^	23.9 ± 1.1 ^a^	8.5 ± 0.6 ^ab^	13.5 ± 1.4 ^ab^	21.4 ± 1.3 ^ab^	18.5 ± 1.0 ^b^	12.0 ± 0.5 ^ab^	9.5 ± 0.6 ^a^	214.1 ± 17.1 ^b^
LSD_0.05_	3.6 ± 1.7	2.0 ± 1.0	1.7 ± 0.8	9.3 ± 4.5	3.4 ± 1.6	2.9 ± 1.4	2.4 ± 1.2	4.0 ± 2.0	2.5 ± 1.2	2.3 ± 1.1	1.5 ± 0.7	43.9 ± 21.3

FSC = fresh seeds that were neither aged nor primed; ASC = seeds that were aged but not primed; FSP.CW = fresh seeds primed with cathodic water; FSP.CM = fresh seeds primed with calcium magnesium solution; FSP.DW = fresh seeds primed with deionized water; ASP.CW = aged seeds primed with cathodic water; ASP.CM = aged seeds primed with calcium magnesium solution; ASP.DW = aged seeds primed with deionized water. ANOVA was performed across treatments. Means of replicate were separated at LSD_0.05_. Post hoc was done using a Tukey test. Means along the same row with different letters were significantly different (*p* < 0.05, *n* = 32).

**Table 3 plants-10-01170-t003:** Effect of cathodic water, calcium magnesium solution and deionized water treatment on the root mass, stem mass, leaves mass, shoot mass, total biomass and shoot/root ratio of *Bolusanthus speciosus, Combretum erythrophyllum* and *Erythrina caffra*.

Dry Mass (g Plant^−1^)		FSC	FSP.CW	FSP.CM	FSP.DW	ASC	ASP.CW	ASP.CM	ASP.DW	LSD_0.05_
Root mass	*B. speciosus*	0.12 ± 0.004 ^cd^	0.20 ± 0.013 ^e^	0.15 ± 0.013 ^de^	0.18 ± 0.015 ^e^	0.04 ± 0.004 ^a^	0.12 ± 0.009 ^bcd^	0.07 ± 0.008 ^ab^	0.10 ± 0.011 ^bc^	0.03 ± 0.01
Stem mass		0.13 ± 0.008 ^b^	0.20 ± 0.003 ^c^	0.13 ± 0.004 ^b^	0.12 ± 0.030 ^b^	0.05 ± 0.001 ^a^	0.11 ± 0.004 ^b^	0.09 ± 0.002 ^ab^	0.10 ± 0.002 ^ab^	0.03 ± 0.02
Leaf mass		0.19 ± 0.006 ^b^	0.28 ± 0.016 ^c^	0.31 ± 0.007 ^c^	0.30 ± 0.036 ^c^	0.10 ± 0.004 ^a^	0.19 ± 0.007 ^b^	0.11 ± 0.003 ^a^	0.16 ± 0.006 ^ab^	0.04 ± 0.02
Shoot mass		0.32 ± 0.012 ^cd^	0.48 ± 0.014 ^e^	0.44 ± 0.009 ^e^	0.42 ± 0.066 ^de^	0.16 ± 0.004 ^a^	0.30 ± 0.009 ^bc^	0.20 ± 0.003 ^ab^	0.26 ± 0.004 ^abc^	0.07 ± 0.03
Total biomass		0.45 ± 0.014 ^c^	0.68 ± 0.023 ^d^	0.59 ± 0.019 ^d^	0.59 ± 0.068 ^d^	0.20 ± 0.007 ^a^	0.41 ± 0.011 ^c^	0.27 ± 0.009 ^ab^	0.37 ± 0.012 ^bc^	0.08 ± 0.04
Shoot/root ratio		2.70 ± 0.091 ^a^	2.44 ± 0.158 ^a^	2.93 ± 0.198 ^a^	2.38 ± 0.371 ^a^	3.60 ± 0.318 ^a^	2.60 ± 0.260 ^a^	2.91 ± 0.357 ^a^	2.69 ± 0.298 ^a^	0.80 ± 0.39
Root mass	*C. erythrophyllum*	0.34 ± 0.035 ^c^	0.68 ± 0.008 ^e^	0.53 ± 0.018 ^d^	0.33 ± 0.031 ^c^	0.08 ± 0.009 ^a^	0.25 ± 0.012 ^c^	0.16 ± 0.009 ^ab^	0.16 ± 0.004 ^b^	0.05 ± 0.03
Stem mass		0.20 ± 0.025 ^b^	0.51 ± 0.041 ^d^	0.39 ± 0.018 ^c^	0.25 ± 0.019 ^b^	0.06 ± 0.004 ^a^	0.19 ± 0.006 ^b^	0.09 ± 0.006 ^a^	0.05 ± 0.003 ^a^	0.06 ± 0.03
Leaf mass		0.35 ± 0.018 ^b^	0.65 ± 0.02 ^c^	0.65 ± 0.016 ^c^	0.37 ± 0.016 ^b^	0.067 ± 0.004 ^a^	0.29 ± 0.032 ^b^	0.13 ± 0.011 ^a^	0.12 ± 0.009 ^a^	0.05 ± 0.02
Shoot mass		0.55 ± 0.027 ^bc^	1.16 ± 0.036 ^e^	1.04 ± 0.022 ^d^	0.62 ± 0.015 ^c^	0.13 ± 0.007 ^a^	0.48 ± 0.034 ^b^	0.23 ± 0.016 ^a^	0.18 ± 0.008 ^a^	0.07 ± 0.03
Total biomass		0.90 ± 0.061 ^cd^	1.84 ± 0.043 ^f^	1.57 ± 0.036 ^e^	0.96 ± 0.039 ^d^	0.20 ± 0.009 ^a^	0.74 ± 0.025 ^c^	0.38 ± 0.025 ^b^	0.34 ± 0.007 ^ab^	0.10 ± 0.05
Shoot/root ratio		1.66 ± 0.096 ^ab^	1.70 ± 0.037 ^ab^	1.95 ± 0.048 ^b^	1.91 ± 0.177 ^b^	1.74 ± 0.328 ^ab^	1.93 ± 0.224 ^b^	1.45 ± 0.024 ^ab^	1.07 ± 0.070 ^a^	0.47 ± 0.23
Root mass	*E. caffra*	0.87 ± 0.106 ^cd^	0.82 ± 0.029 ^bcd^	0.85 ± 0.072 ^cd^	1.08 ± 0.153 ^d^	0.26 ± 0.025 ^a^	0.73 ± 0.035 ^bcd^	0.55 ± 0.085 ^abc^	0.46 ± 0.058 ^ab^	0.24 ± 0.12
Stem mass		1.06 ± 0.074 ^bcd^	1.56 ± 0.149 ^d^	1.30 ± 0.181 ^cd^	1.37 ± 0.144 ^cd^	0.20 ± 0.038 ^a^	0.93 ± 0.170 ^bc^	0.27 ± 0.038 ^a^	0.56 ± 0.069 ^ab^	0.35 ± 0.17
Leaf mass		2.02 ± 0.175 ^bc^	3.31 ± 0.202 ^d^	2.26 ± 0.163 ^bc^	2.70 ± 0.302 ^cd^	0.21 ± 0.041 ^a^	1.46 ± 0.247 ^a^	0.42 ± 0.064 ^b^	0.58 ± 0.075 ^a^	0.53 ± 0.26
Shoot mass		3.08 ± 0.208 ^bc^	4.88 ± 0.306 ^d^	3.56 ± 0.308 ^bc^	4.07 ± 0.434 ^cd^	0.41 ± 0.068 ^a^	2.39 ± 0.292 ^b^	0.69 ± 0.101 ^a^	1.15 ± 0.140 ^a^	0.76 ± 0.37
Total biomass		3.95 ± 0.225 ^bc^	5.70 ± 0.278 ^d^	4.41 ± 0.355 ^bcd^	5.15 ± 0.579 ^cd^	0.68 ± 0.089 ^a^	3.12 ± 0.274 ^b^	1.24 ± 0.159 ^a^	1.61 ± 0.197 ^a^	0.88 ± 0.43
Shoot/root ratio		3.75 ± 0.596 ^c^	6.01 ± 0.594 ^d^	4.20 ± 0.293 ^cd^	3.83 ± 0.245 ^c^	1.57 ± 0.176 ^ab^	3.34 ± 0.524 ^bc^	1.32 ± 0.190 ^a^	2.50 ± 0.086 ^abc^	1.13 ± 0.55

Abbreviations as for Table 2. ANOVA was performed across treatments. The means of replicate was separated at LSD_0.05_. Post hoc was done using a Tukey test. Means along the same row with different letters were significantly different (*p* < 0.05, *n* = 32).

## Data Availability

Data is contained within the article.

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
