# Peer review of "Cathodic Water Enhances Seedling Emergence and Growth of Controlled Deteriorated Orthodox Seeds"

_plants, 2021, doi:10.3390/plants10061170_

Round 1
Reviewer 1 Report
Dear Authors,
Manuscript titled: “CATHODIC WATER ENHANCES SEEDLING EMERGENCE 2 AND GROWTH OF CONTROLLED DETERIORATED OR-3 THODOX SEEDS.” addresses an important issue of studying the effect of priming to ameliorate deterioration of seeds. I think that this very interesting topic which is insufficiently explored in scientific literature. I agree with authors, that the deterioration of seeds in seed banks is of global concern, and we should put more attention to find why how to slow down the deterioration of seeds during storage. Therefore, I think that this article could be interesting for broad scientific community related to seed science. The presented manuscript is consistent and well written. The statements are clear. The results are presented in a transparent and clear manner. Discussion chapter could be improved by putting more attention on the explanation how cathodic water can affect the level of 4-HNE in seeds. Please find below also the list of my concerns which needs explanation.
Considering the entire manuscript, the quality and presentation of the results in my opinion this article can be published in Plants after minor revision.
Table 1
Is it correct “Means along the same row with different letters were significantly different (p0.05, n=32)”? Or should it be Means along the same line with different letters were significantly different (p0.05, n=32)? I believe it was not an intention of author to compare the data of First day of emergence, Emergence %, Mean emergence time etc. among one treatment but rather between them.
Please check and correct the letters showing significantly different in the case of First day of emergence for Bolusanthus speciosus (Table 1). For me is in unclear why there is no significant difference between first day of emergence for FSP.CW (4.5 days) and for ASC (9.5 days), both are marked with the same letter “a” ,but ASP.DW (8.5 days) and ASP.CM (8.0 days) had significantly delayed first day of emergence as they are marked “cd” and “bcd”, respectively . I assume there is a mistake in presentation of statistic results. Please check the accuracy.
Correct number of table instead “Table 6. 2: Effects of cathodic water, calcium magnesium solution and deionized water treatments on the root length, stem length, number of leaves and leaf area of Bolusanthus speciosus, Combretum erythrophyllum and Erythrina caffra.”
It should be:
Table 2: Effects of cathodic water, calcium magnesium solution and deionized water treatments on the root length, stem length, number of leaves and leaf area of Bolusanthus speciosus, Combretum erythrophyllum and Erythrina caffra.”
Table 3
“Means along the same row with different letters were significantly different (p0.05, n=32).” Correct to “ Means along the same line with different letters were significantly different”
Effects on membrane leakage and lipid peroxidation products
“However, priming only reduced levels of the other oxidation product we tested, 4- HNE, in E. caffra, and all solutions were equally effective.”
According to Figure 4 in the case of Bolusanthus speciosus and Combretum erythrophyllum both cathodic water (ASP.CW) and calcium magnesium (ASP.CM) slightly reduced detected level of 4-HNE in aged seeds. Moreover, the observed reduced levels of 4-HNE did not differ significantly from control samples (FSC). This observation should be also mentioned and discussed in manuscript.
Author Response
Dear Reviewer,
We appreciate your comments and suggestions and we have attended to them. Your scrutiny/ review has no doubt improved our manuscripts.
Thanks
Kayode
REVIEWER 1
Comments and Suggestions for Authors
COMMENT Table 1: Is it correct “Means along the same row with different letters were significantly different (p<0.05, n=32)”? Or should it be Means along the same line with different letters were significantly different (p<0.05, n=32)? I believe it was not an intention of author to compare the data of First day of emergence, Emergence %, Mean emergence time etc. among one treatment but rather between them…. |
RESPONSE The use of ‘line’ may be confusing, a line could mean a column; row or it could be diagonal. Hence the use of row was not changed in the m/s |
COMMENT Please check and correct the letters showing significantly different in the case of First day of emergence for Bolusanthus speciosus (Table 1). For me is in unclear why there is no significant difference between first day of emergence for FSP.CW (4.5 days) and for ASC (9.5 days), both are marked with the same letter “a” ,but ASP.DW (8.5 days) and ASP.CM (8.0 days) had significantly delayed first day of emergence as they are marked “cd” and “bcd”, respectively . I assume there is a mistake in presentation of statistic results. Please check the accuracy |
RESPONSE There was a typographical error, it ought to be ‘d’, it has been corrected in the table We really appreciate this your observation which has so far escaped our scrutiny |
COMMENT Correct number of table instead “Table 6. 2: Effects of cathodic water, calcium magnesium solution and deionized water treatments on the root length, stem length, number of leaves and leaf area of Bolusanthus speciosus, Combretum erythrophyllum and Erythrina caffra.” |
RESPONSE It should be: Table 2…it has been corrected thus: Table 2: Effects of cathodic water, calcium magnesium solution and deionized water treatments on the root length, stem length, number of leaves and leaf area of Bolusanthus speciosus, Combretum erythrophyllum and Erythrina caffra.” |
COMMENT Table 3 “Means along the same row with different letters were significantly different (p<0.05, n=32).” Correct to “ Means along the same line with different letters were significantly different” |
RESPONSE The use of line may be confusing, a line could mean a column, row or it could be diagonal. Hence the use of row was not changed in the m/s |
COMMENT According to Figure 4 in the case of Bolusanthus speciosus and Combretum erythrophyllum both cathodic water (ASP.CW) and calcium magnesium (ASP.CM) slightly reduced detected level of 4-HNE in aged seeds. Moreover, the observed reduced levels of 4-HNE did not differ significantly from control samples (FSC). This observation should be also mentioned and discussed in manuscript.
|
RESPONSE The observation has been mentioned in the manuscripts as recommended |
NOTE: All other minor suggestions have been attended to
Reviewer 2 Report
This is a very interesting paper that will be of benefit to all seed banks engaged in long-term seed storage. Corrections and suggested amendments are in the attached file; these relate mainly to clarification of methods and improved presentation of results. I look forward to seeing this work published.

Author Response
Dear reviewer,
We have attended to all your comments and suggestions point by point. Your contributions have no doubt improve our manuscripts.
Thank you,
Kayode
REVIEWER 2
Comments and Suggestions for Authors
COMMENTS |
RESPONSES
|
Aggravated temperature |
Aggravated can not be left out. The temperature has to be increased/ aggravated before controlled deterioration can take place, otherwise it will be natural deterioration
|
Replace school with research unit |
Here, the green houses do not belong to our research unit. They belong to the school of life sciences. We share with other research units in the school
|
Did you conduct imbibition test? |
Imbibition test was conducted. The statement has been restructured
|
Was pH tested?
|
Yes
|
…..water containing CaMg solution
|
It has been corrected to calcium magnesium solution; Water has been removed
|
An agar-based salt bridge
|
An agar-based salt bridge (U shaped glass tube filled with electrolyte) – The statement has been restructured to make it clearer
|
Composition of Grovida potting mix |
Grovida potting mix is made of peat - Changed to peat based Grovida potting mix
|
How often was fertilizer applied?
|
Weekly - Weekly, Grovida multifeed water soluble fertilizer was used to supply nutrients to the growing plants at 1 g l-1.
|
TBA
|
TBA has been defined – thiobarbituric acid
|
About the end of phase II of the seed hydration
|
About the end of phase II of the seed hydration – Duration (hour) has been included
|
Reviewer 3 Report
The authors investigated germination and growth of three different South African tree species that produce orthodox seeds. Improving germination after long-term storage of seeds is important regarding the conservation of plant species. The authors show that treatment of artificial deteriorated seeds with cathodic water can ameliorate deterioration. To my opinion, the manuscript can be published after minor revision. The authors should comment on the following:
How was the initial water content of the seeds raised to 14%? How was this measured? (p. 3, l. 28)
The three species display different quantitative effects on seedling growth after priming (chapter 3.2). Especially for E. caffra, priming with cathodic water had a significant positive effect (p.10, l. 6). Can this be explained regarding seed/plant morphology/physiology?
The authors observe differences regarding membrane leakage and lipid peroxidation products between species. The author should discuss these differences in more detail (p. 15 l.19-27). Can these differences be explained?
Minor comments:
p.3,l. 32: "A solution containing 1µM CaCl2 and 1 µM MgCl2 in deionized water (?)..."
p.3., l. 36: "Two 200 ml glass beakers were filled with CaMg solution (?)..."
p.3. l. 40: "...at 400 mA..."
p.3 l. 41+42: What does the abbreviation c. mean?
chapter 2.2: It is not fully clear how many seeds were used for the experiments. The authors describe that 50 seeds were used per treatment, and that each treatment was replicated four times. So 200 seeds were used altogether per treatment? To test emergence and subsequent seedling growth, five seeds were planted per pot, with four pots per treatment. So 20 seeds were used per treatment altogether?
p.4, l.17: WAP = weeks after planting?
p.5, l.6: µmol m-2 s-1
p.5 l. 22: Explain the abbreviation TBA.
p.5 l. 48: Check formula.
p.14 l39-40: Sentence is unclear.
Author Response
Dear Reviewer,
We thank you for the thorough review of our manuscript which you did. Without any doubt, your comments and suggestions have really improved our m/s. Efforts were made to attend to all your comments.
Thanks a lot,
Kayode FATOKUN
REVIEWER 3
Comments and Suggestions for Authors
COMMENTS
|
RESPONSES |
How was the initial water content of the seeds raised to 14%? How was this measured? (p. 3, l. 28)
|
The water content for all the species was raised to 14% using a vapour chamber. The seeds were then sealed in airtight glass jars and kept in a digital oven (Series 2000, Scientific, USA) at 40°C and 100% relative humidity.
|
The three species display different quantitative effects on seedling growth after priming (chapter 3.2). Especially for E. caffra, priming with cathodic water had a significant positive effect (p.10, l. 6). Can this be explained regarding seed/plant morphology/physiology?
|
It has been explained The improvement in seedling growth in plants derived from primed seeds may have occurred due to increase in the activities of enzymes such as α-amylase. Such increases in enzymatic activities have been reported to promote the hydrolysis of starch into soluble sugars for seed respiration and better growth [22].
|
The authors observe differences regarding membrane leakage and lipid peroxidation products between species. The author should discuss these differences in more detail (p. 15 l.19-27). Can these differences be explained?
|
Discussion has been enhanced. However, the research the did not cover investigating the biological properties of the each species that may have been responsible for the differences in their responses to controlled deterioration and priming |
p.3,l. 32: "A solution containing 1µM CaCl2 and 1 µM MgCl2 in deionized water (?)..."
|
corrected
|
p.3., l. 36: "Two 200 ml glass beakers were filled with CaMg solution (?)..."
|
corrected |
p.3. l. 40: "...at 400 ma..."
|
corrected….400mA |
p.3 l. 41+42: What does the abbreviation c. mean?
|
c. means approximately…it has been explained |
chapter 2.2: It is not fully clear how many seeds were used for the experiments. The authors describe that 50 seeds were used per treatment, and that each treatment was replicated four times. So 200 seeds were used altogether per treatment? To test emergence and subsequent seedling growth, five seeds were planted per pot, with four pots per treatment. So 20 seeds were used per treatment altogether?
|
200 seeds were used per treatment [… Each treatment was replicated four times (n=200). Note: n = total number of seeds used per treatment.
To test emergence and subsequent seedling growth 20 seeds were used per treatment. [….After CD and priming, five seeds were planted in each pot, with four pots per treatment (n=20) arranged in a completely randomized design]. |
p.4, l.17: WAP = weeks after planting?
|
The abbreviation has been defined at first mention: 8 weeks after planting (WAP),
|
p.5, l.6: µmol m-2 s-1 |
Corrected: Instantaneous measurement of leaf based CO2 assimilation and transpiration rates were carried out at a CO2 concentration of 400 µmol CO2 mol-1, a light intensity of 1000 µmol m-2 s-1 |
p.5 l. 22: Explain the abbreviation TBA. |
It has been explained as recommended: Thiobarbituric acid
|
p.5 l. 48: Check formula. |
Corrected …..MSI = [1- (T1/T2)] X 100
|
p.14 l39-40: Sentence is unclear. |
Sentence has been improved on
|